# Amorphous Sb_2_S_3_ Nanospheres In-Situ Grown on Carbon Nanotubes: Anodes for NIBs and KIBs

**DOI:** 10.3390/nano9091323

**Published:** 2019-09-15

**Authors:** Meng Li, Fengbin Huang, Jin Pan, Luoyang Li, Yifan Zhang, Qingrong Yao, Huaiying Zhou, Jianqiu Deng

**Affiliations:** School of Materials Science and Engineering & Guangxi Key Laboratory of Information Materials, Guilin University of Electronic Technology, Guilin 541004, China; muzi_lemon@163.com (M.L.); huang2664110462@163.com (F.H.); shengbin3@163.com (J.P.); lly15677300606@163.com (L.L.); 18174198852@163.com (Y.Z.); qingry96@guet.edu.cn (Q.Y.); zhy@guet.edu.cn (H.Z.)

**Keywords:** Amorphous Sb_2_S_3_/CNT nanocomposites, High capacity, Anode, Na-ion batteries, K-ion batteries

## Abstract

Antimony sulfide (Sb_2_S_3_) with a high theoretical capacity is considered as a promising candidate for Na-ion batteries (NIBs) and K-ion batteries (KIBs). However, its poor electrochemical activity and structural stability are the main issues to be solved. Herein, amorphous Sb_2_S_3_ nanospheres/carbon nanotube (Sb_2_S_3_/CNT) nanocomposites are successfully synthesized via one step self-assembly method. In-situ growth of amorphous Sb_2_S_3_ nanospheres on the CNTs is confirmed by X-ray diffraction, field-emission scanning electron microscopy, and transmission electron microscopy. The amorphous Sb_2_S_3_/CNT nanocomposites as an anode for NIBs exhibit excellent electrochemical performance, delivering a high charge capacity of 870 mA h g^−1^ at 100 mA g^−1^, with an initial coulomb efficiency of 77.8%. Even at 3000 mA g^−1^, a charge capacity of 474 mA h g^−1^ can be achieved. As an anode for KIBs, the amorphous Sb_2_S_3_/CNT nanocomposites also demonstrate a high charge capacity of 451 mA h g^−1^ at 25 mA g^−1^. The remarkable performance of the amorphous Sb_2_S_3_/CNT nanocomposites is attributed to the synergic effects of the amorphous Sb_2_S_3_ nanospheres and 3D porous conductive network constructed by the CNTs.

## 1. Introduction

Recently, Na-ion batteries (NIBs) and K-ion batteries (KIBs) are being developed hotly as potential substitutes for Li-ion batteries (LIBs) in large-scale energy storage system, due to their natural abundance and low cost [1,2]. Owing to the large radius of Na ions (1.02 Å) and K ions (1.38 Å), however, it is difficult to find appropriate electrode materials, as they would need high capacities, rapid diffusion kinetics and long life cycles [3,4]. Judging from the present development of NIBs and KIBs, the research focused on cathode materials is more productive and extensive, such as layered transition metal oxides [5,6], polyanionic type compounds [7,8] and Prussian blue analogs [9,10]. For the anode materials, the available choices are limited, including the carbonaceous materials [11,12], alloys/metals [13,14], metal sulfides [15,16] and phosphides [17,18]. Among these materials, Sb_2_S_3_ is considered as a promising anode material for NIBs and KIBs because of its high theoretical specific capacity, and proper sodium/potassium inserting potential based on the conversion reaction and alloying reaction [19,20,21]. However, the insertion/extraction process of Na/K-ion in Sb_2_S_3_ is accompanied with large volume change and structure destruction, along with its lower conductivity, leading to the poor rate capability and cycling performance [20,22,23].

To address these problems, combining nanostructured Sb_2_S_3_ with carbon materials is regarded as an effective strategy [24,25]. For example, Zhao and Manthiram have reported that an amorphous Sb_2_S_3_-graphite electrode for NIBs delivers a high rate capacity, a high initial oulombic efficiency and stable cycling performance, owing to the amorphous structure of Sb_2_S_3_ and the conductive graphite matrix [26]. Nanostructured Sb_2_S_3_/sulfur-doped graphene (Sb_2_S_3_/SGS) anodes for NIBs demonstrate a stable capacity retention of 83% for 900 cycles, with high capacities of 2 A g^−1^ and excellent rate performances [27]. A remarkable progress on the exploration of Sb_2_S_3_-based anodes for NIBs has been made in the past decade. To our knowledge, however, only a few studies have been reported on the Sb_2_S_3_-based anodes for KIBs, including Sb_2_S_3_@polypyrrole (Sb_2_S_3_@PPy) coaxial nanorods [21], Sb_2_S_3_ nanoparticles (~20 nm) uniformly dispersed into a porous S,N-codoped graphene framework (Sb_2_S_3_-SNG) composite [23], and a multi-layered Sb_2_S_3_/carbon sheet (SBS/C) composite [24]. Many studies are needed to develop Sb_2_S_3_-based electrodes for KIBs with high capacities, enhanced rate capabilities and cycling performances. Herein, we demonstrate the feasibility of amorphous Sb_2_S_3_ nanospheres/carbon nanotube (Sb_2_S_3_/CNT) nanocomposites as anode materials for NIBs and KIBs. Electrochemical measurements indicate the high reversible capacities and superior rate performance of Sb_2_S_3_/CNT nanocomposites, owing to the fact that the 3D porous nanostructure of the composites can accommodate the large volume change and enhance the ionic diffusion kinetics in Sb_2_S_3_.

## 2. Materials and Methods

### 2.1. Materials Synthesis

The amorphous Sb_2_S_3_/CNT nanocomposites were simply synthesized by a one-step self-assembly method reported by Yang’s group with minor modifications [28]. Firstly, 0.34 g SbCl_3_ (99.9%, Alfa Aesar Chemicals Co., Ltd., Shanghai, CN) was dissolved in 30 mL ethylene glycol (EG, 99%, Merck Life Science (Shanghai) Co., Ltd. Shanghai, CN) in a capped glass bottle with magnetic stirring. Secondly, 80 mg CNTs were added in the solution when it turned colorless and was stirred for 3 min to form a black suspension. The CNTs were purchased from XFNANO Company (Nanjing, China). 0.23 g of thioacetamide (TAA, 99%, Merck Life Science (Shanghai) Co., Ltd. Shanghai, China) was then added to the suspension without stirring, and the glass bottle was sealed by tightening the cap. After standing for 24 h at room temperature, red brown powders were formed, and finally, the products were obtained by centrifugation, washing several times with deionized water and ethanol, and drying in a vacuum oven at 90 °C for 24 h. Amorphous Sb_2_S_3_ nanospheres were also prepared by the same procedure without the CNTs.

### 2.2. Materials Characterization

Powder X-ray diffraction (XRD) measurements were performed on PIXcel^3D^ X-ray diffractometer using Cu *K*_α_ radiation source to identify the crystal structure of the materials. The morphology of the products was analyzed by field emission scanning electron microscopy (FESEM, Hitachi SU-70, Hitachi High-Technologies Corporation, Tokyo, Japan). Transmission electron microscopy (TEM) and high-resolution TEM (HRTEM) images were recorded by using a transmission electron microscope (Tecnai G2 F20, FEI Company, Hillsboro, Oregon, USA). The content of the CNTs was evaluated by an acid dissolution method. Typically, 3.0 g nanocomposites were dissolved in 50 mL concentrated hydrochloric acid (6 M). The CNTs were collected by centrifugation and washed several times with deionized water and ethanol, and then dried in a vacuum oven at 90 °C for 24 h. The content was determined according to the weight of residual CNTs.

### 2.3. Electrochemical Measurements

Electrochemical measurements were conducted in 2032-type coin cells, which were assembled in a high-purity argon-filled glove box. The amorphous Sb_2_S_3_/CNT nanocomposites were mixed with acetylene black, and carboxy methyl cellulose (CMC) with a ratio of 7:2:1 in deionized water to form slurry. The slurry was pasted on a Cu foil and dried at 80 °C under vacuum for 12 h. The loading density of the electrode laminates was about 1.5 mg cm^−2^. Sodium and potassium foils were used as counter and reference electrodes, respectively. The electrolyte solution was 1 M NaClO_4_ and 0.8 M KPF_6_ dissolved in ethylene carbonate and diethyl carbonate (1:1 by weight) for NIBs and KIBs, respectively. The separator was the Celgard 2400 membrane. Galvanostatic discharge-charge tests were performed by an Arbin battery testing system (BT-2000). The capacities of the amorphous Sb_2_S_3_/CNT anode were determined on the basis of the total mass of Sb_2_S_3_ and CNTs. Electrochemical impedance spectroscopy (EIS) and cyclic voltammetry (CV) measurements were carried out on a Solartron electrochemical workstation.

## 3. Results and Discussion

### 3.1. Materials Structure and Morphology

The detailed structural and morphological characterizations of the Sb_2_S_3_/CNT nanocomposites are shown in Figure 1 and Figure 2. As shown in Figure 1, no obvious sharp diffraction peaks are observed in the XRD pattern of pure Sb_2_S_3_, indicating the amorphous structure of as-prepared Sb_2_S_3_. For the Sb_2_S_3_/CNT nanocomposites, the XRD pattern overlaps well with that of pure Sb_2_S_3_, except a slightly sharp peak at 2*θ* = 26° arising from the strongest peak of the CNTs. The results imply that Sb_2_S_3_ in the nanocomposites is amorphous, and the presence of the CNTs cannot affect the amorphous structure of Sb_2_S_3_. As can be seen from Figure 2a,b, the amorphous Sb_2_S_3_ nanospheres grow uniformly on the CNTs. The amorphous Sb_2_S_3_/CNT nanocomposites interlink and interweave to form a 3D porous conductive network. The amorphous Sb_2_S_3_/CNT nanocomposites have a diameter range of 30–150 nm, which results from the recombination of pure Sb_2_S_3_ nanospheres and the CNTs. The grain size of pure Sb_2_S_3_ micro/nanospheres is in a range of 50–1200 nm, as displayed in Appendix A. The presence of the CNTs inhibits the growth of Sb_2_S_3_ without changing the spherical morphology. TEM and HRTEM were performed to investigate the microstructure of the CNTs and amorphous Sb_2_S_3_/CNT nanocomposites. From TEM image (Appendix A), the CNTs show a wire structure and interconnect to one another. The diameters are in range of 20–90 nm. HRTEM image (Appendix A) reveals the amorphous carbon microstructure of the CNTs that are composed of short-range ordered nano-crystallites. TEM images (Figure 2c,d) further reveal that the Sb_2_S_3_ nanospheres with an average particle size of 80 nm are well grown on the CNTs. From the illustration in Figure 2d, the diffuse rings in the selected-area electron diffraction (SEAD) pattern confirm the amorphous structure of Sb_2_S_3_. Furthermore, the EDX spectrum in Figure 2e undoubtedly proves the nanocomposites contain element Sb, S and C. As expected, the atomic ratio of S and Sb is 1.53. The signals of Cu and O peaks originate from the carbon-coated copper grid. Additionally, the CNTs content is about 21 wt.%, which was determined by using the acid dissolution method.

### 3.2. Amorphous Sb_2_S_3_/CNT Nanocomposites as an Anode for NIBs

Figure 3a displays the CV curves of the amorphous Sb_2_S_3_/CNT nanocomposites as an anode for NIBs during the initial five cycles at a scan rate of 0.05 mV s^−1^ between 0.01 and 1.5 V (versus Na/Na^+^). The first cathodic scan displays two strong current peaks located at 0.83 and 0.41 V, respectively, corresponding to the conversion reaction and alloying reaction of Sb to Na_3_Sb (Equations (1) and (2)) [15]. Two oxidation peaks located at 0.8 and 1.3 V in the first anodic scan are assigned to the dealloying and reconversion reactions [26]. A pair of small redox peaks near 0.01 V in the CV curves is assigned to the Na-ion insertion/extraction reaction in the conductive carbon materials (acetylene black) and the CNTs [25]. The subsequent CV curves are almost overlapping, especially the pair of redox peaks of alloying/dealloying reaction, indicating an excellent cycle stability of the Sb_2_S_3_/CNT anodes.
Sb_2_S_3_ + 6Na^+^ + 6e^−^ → 2Sb + 3Na_2_S(1)
2Sb + 6Na^+^ + 6e^−^ → 2Na_3_Sb(2)

The discharge/charge plateau regions can be clearly observed in the discharge-charge profiles of the Sb_2_S_3_/CNT anode under the current density of 100 mA g^−1^ between 0.01 and 1.5 V (Figure 3b), in accordance with the CV curves. The capacities of the Sb_2_S_3_/CNT anode were calculated on the basis of the total mass of Sb_2_S_3_ and CNTs. The Sb_2_S_3_/CNT anode delivers a high initial, reversible charge capacity of 870 mA h g^−1^ and a coulomb efficiency of 77.8%. The initial discharge capacity of 1130 mA h g^−1^ is higher than the theoretical capacity (946 mA h g^−1^), which can be mainly ascribed to the formation of solid electrolyte interface (SEI) film and the decomposition of the electrolyte. From the Figure 3c, the Sb_2_S_3_/CNT anode exhibits superior cycling performance at 100 mA g^−1^, delivering a charge capacity of 704 mA h g^−1^ after 50 cycles, corresponds to 81% of the initial charge capacity. The cycling performance is much higher than that of pure Sb_2_S_3_ with a reversible capacity of 262 mA h g^−1^ over 25 cycles (Appendix A).

The rate capability of the Sb_2_S_3_/CNT anode at various current densities was further studied, as show in Figure 3d and Appendix A. The discharge capacities of 1280, 831, 795, 743, 685, 601 and 474 mA h g^−1^ can be achieved by cycling at various current densities of 50, 100, 200, 500, 1000, 2000 and 3000 mA g^−1^. The corresponding charge capacities are 874, 813, 749, 725, 633, 576 and 441 mA h g^−1^. After 35 discharge-charge cycles, the Sb_2_S_3_/CNT anode can maintain a discharge capacity of 791 mA h g^−1^ when the current density returns to 50 mA g^−1^, indicating excellent rate performance. To highlight the superior electrochemical properties of the Sb_2_S_3_/CNT anode, a comparison of the electrochemical performance of the amorphous Sb_2_S_3_/CNT nanocomposites with that of previously reported Sb_2_S_3_-based anode materials for NIBs is shown in Table 1. As can be seen, the Sb_2_S_3_/CNT anode is comparable with the reported Sb_2_S_3_@PPy [21], MWNTs@Sb_2_S_3_@PPy [25], Sb_2_S_3_/graphene [29], hollow-sphere (HS) Sb_2_S_3_/C [30], Sb_2_S_3_/PPy [31], sulfur and nitrogen dual doped reduced graphene oxide/Sb_2_S_3_ (SN-rGO/Sb_2_S_3_) [32], stibnite/sulfur-doped carbon sheet (Sb_2_S_3_/SCS) [33], hierarchical Sb_2_S_3_ hollow microspheres (Sb_2_S_3_ HMS) [34] and RGO/Sb_2_S_3_ nanorods [35] in terms of initial coulomb efficiency, reversible specific capacity and rate performance. That is attributed to the synergic effects of the amorphous Sb_2_S_3_ nanospheres and the incorporation of the CNTs. The intrinsic isotropic nature of the amorphous Sb_2_S_3_ can accommodate the volume change and keep the electrical contact during discharge/charge cycling, particularly in the first cycling, thus facilitating the high initial coulomb efficiency and capacity [19,26]. No obvious morphology differences of the pristine and cycled amorphous Sb_2_S_3_-graphite electrodes have been observed by using SEM [26], confirming the structural stability of amorphous Sb_2_S_3_ during cycling. The amorphous Sb_2_S_3_ nanospheres can provide more active sites and shorten the diffusion pathway of sodium ion during sodiation/desodiation process, being beneficial to the high capacity and superior rate performance. The 3D porous conductive network composed of the CNTs can well buffer the volume change of Sb_2_S_3_ and maintain the structural integrity of the electrode [22,25], consequently, leading to a considerable electrochemical performance.

### 3.3. Amorphous Sb_2_S_3_/CNT Nanocomposites as an Anode for KIBs

Inspired by the excellent electrochemical characteristics of the amorphous Sb_2_S_3_/CNT nanocomposites as an anode of NIBs, we further studied their electrochemical performance in KIBs. The potassiation–depotassiation behavior of the Sb_2_S_3_/CNT anode was firstly investigated by CV technique at a scan rate of 0.05 mV s^−1^ in a range of 0.01 to 3.0 V (versus K/K^+^), as shown in Figure 4a. The first CV curve reveals three obvious redox peaks located at 1.35–2.20 V, 0.80–1.37 V and 0.35–1.0 V, which are ascribed to the conversion and alloying reactions during potassiation–depotassiation process [21,23,24]. A small cathodic peak at 1.78 V may be related to the SEI film formation and the intercalation of K-ion into the Sb_2_S_3_ layered structure. For the subsequent scans, the intensity and location of redox peaks gradually alter, which is due to the structural change of Sb_2_S_3_ caused by the K-ion insertion/extraction. The reversible potassiation–depotassiation reactions of the Sb_2_S_3_/CNT anode can be described as in Equations 3–5, which has been confirmed by Lu et al. [23] and Liu et al. [24].
**Intercalation reaction:** Sb_2_S_3_ + *x*K^+^ + *x*e^−^ → K*_x_*Sb_2_S_3_(3)
**Conversion reaction:** K*_x_*Sb_2_S_3_ + 2*x*K^+^ + 2*x*e^−^ → 2Sb + K*_x_*S(4)
**Alloying reaction:** 2Sb + K*_x_*S + (8 − 3*x*)K^+^ + (8 − 3*x*) e^−^ → 2K_3_Sb_2_ + K_2_S_3_(5)

The galvanostatic discharge/charge curves of the Sb_2_S_3_/CNT anode measured at various current densities in a range of 0.01–2.5V are shown in the Figure 4b. For the first discharge-charge curves at 25 mA g^−1^, two pairs of obvious discharge/charge plateaus at about 1.1–1.3 V and at 0.5–0.6 V can be observed in the curves, in accordance with the CV curves. In addition, discharge sloping regions from 0.4 to 0.01 V presents in the discharge curve. The results indicate the multistep potassiation–depotassiation processes, including the intercalation reaction, the conversion reaction and alloying reaction. The discharge and charge capacities are 869 and 451 mA h g^−1^, respectively. The corresponding coulomb efficiency is 52%. The capacity loss is mainly caused by the SEI formation and irreversible side reaction [23]. The charge capacity of the Sb_2_S_3_/CNT anode is higher than that of the developed anode materials, such as SnS_2_-rGO (355 mA h g^−1^ at 25 mA g^−1^) [16], hard carbon microspheres (262 mA h g^−1^ at 28 mA g^−1^) [38], Sn–C composite (140 mA h g^−1^ at 25 mA g^−1^) [39], and MoS_2_ (65.4 mA h g^−1^ at 20 mA g^−1^) [40]. The Sb_2_S_3_/CNT anode delivers the charge capacities of 446.8, 254.7, 222.2 and 166.6 mA h g^−1^ at different current densities of 50, 100, 500 and 1000 mA g^−1^ (Figure 4c). The cycling performance of the Sb_2_S_3_/CNT anode at 500 mA g^−1^ is shown in Figure 4d. The first discharge and charge capacities are 691 and 286.5 mA h g^−1^, respectively, corresponding to an initial coulomb efficiency of 41.5%. After 50 cycles, the reversible capacity is 212.4 mA h g^−1^, corresponding to a capacity retention of 74.2%. Except the first cycle, the coulomb efficiency is above 95% during the whole cycling. The inferior electrochemical performance of the amorphous Sb_2_S_3_/CNT nanocomposites in KIBs compared to that in NIBs could be attributed to the large volume expansion, severe pulverization and aggregation, and slower diffusion kinetics of K-ions during the repeated potassiation–depotassiation processes, resulting from the larger ionic radius of the K-ion (1.38 Å) [21,41]. Similar results have also been reported in VS_2_ nanosheet [42] and ReS_2_ nanosheets [43]. Additionally, compared to that of the previously reported Sb_2_S_3_-based anodes for KIBs (Table 2), the performance of amorphous Sb_2_S_3_/CNT nanocomposites in this work is poor, indicating the Sb_2_S_3_/CNT could not deliver their capacities sufficiently during the cycling. The direct growth of Sb_2_S_3_ nanospheres on the CNTs cannot effectively buffer the volume expansion/contraction and release the mechanical stress of Sb_2_S_3_, leading to irreversible potassiation–depotassiation reactions. Those issues can be resolved by surface coating effective conducting agents, or by fabricating nanocomposites with unique morphologies, such as nanosheets, nanotubes and hollow nanospheres.

## 4. Conclusions

In summary, the amorphous Sb_2_S_3_/CNT nanocomposites are successfully prepared via in-situ growth of the amorphous Sb_2_S_3_ nanospheres on the CNTs, constructing a unique nanostructure. The amorphous Sb_2_S_3_/CNT nanocomposites as an anode for NIBs demonstrate an attractive reversible charge capacity of 870 mA h g^−1^ and a high initial coulomb efficiency of 77.8% at 100 mA g^−1^. The stable charge capacity is maintained at 704 mA h g^−1^ over 50 cycles. Even at a high current density of 3000 mA g^−1^, a charge capacity of 474 mA h g^−1^ can be achieved. As an anode for KIBs, the amorphous Sb_2_S_3_/CNT nanocomposites also demonstrate a high charge capacity of 451 mA h g^−1^ at 25 mA g^−1^, exceeding the capacities of hard carbon, Sn–C composite and SnS_2_-rGO; but future studies are required to optimize the rate performance and cycling stability. The remarkable performance of the amorphous Sb_2_S_3_/CNT nanocomposites in NIBs is attributed to the synergic effects of the amorphous Sb_2_S_3_ nanospheres and 3D porous conductive network composed of the CNTs.

## Figures and Tables

**Figure 1 nanomaterials-09-01323-f001:**
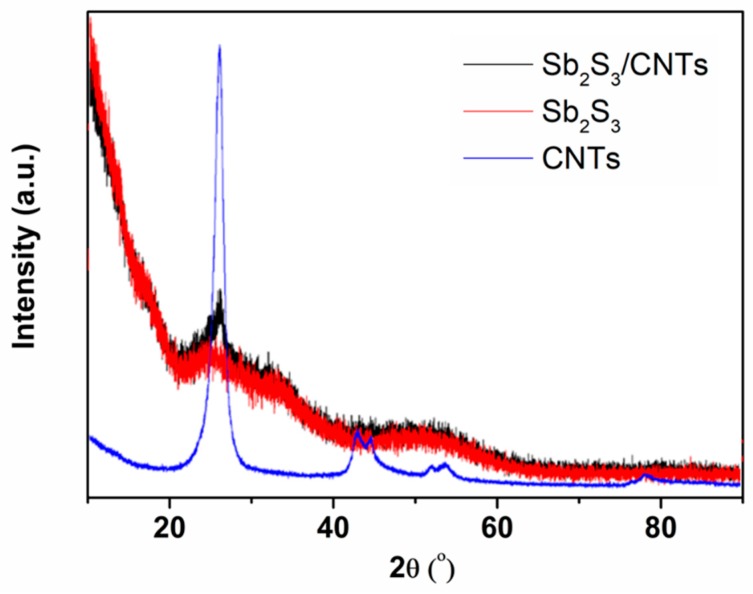
XRD patterns of the as-prepared Sb_2_S_3_/carbon nanotube (CNT) nanocomposites, pure Sb_2_S_3_ and CNTs.

**Figure 2 nanomaterials-09-01323-f002:**
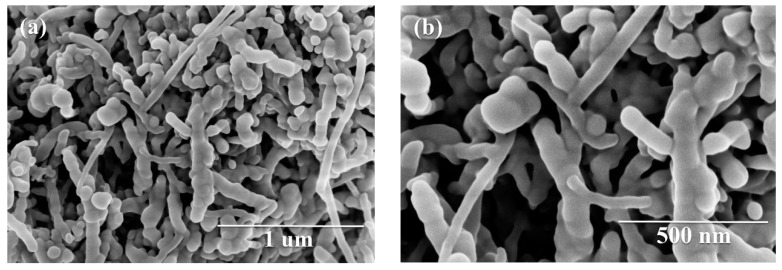
SEM and TEM images of the as-synthesized Sb_2_S_3_/CNT nanocomposites. (**a**,**b**) SEM images, (**c**,**d**) TEM images and (**e**) EDX spectrum. Selected-area electron diffraction (SAED) pattern is illustrated in (**d**).

**Figure 3 nanomaterials-09-01323-f003:**
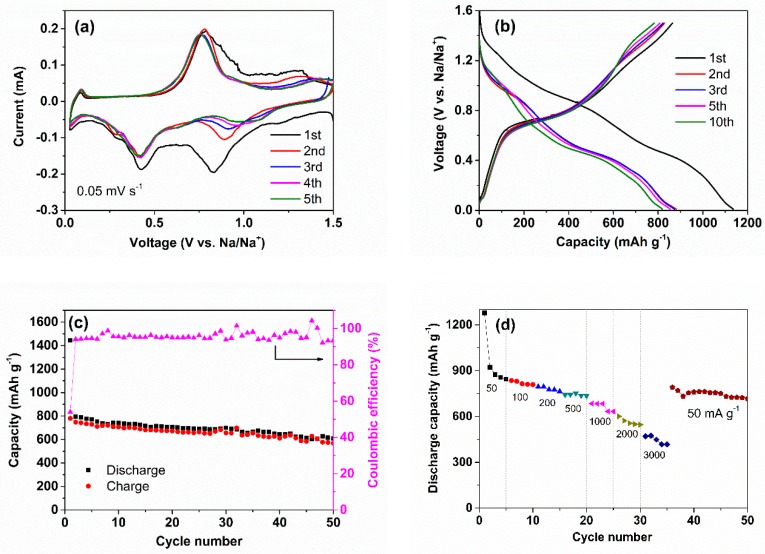
Electrochemical properties of the Sb_2_S_3_/CNT anode for Na-ion batteries (NIBs) in a voltage range of 0.01–1.5 V. (**a**) Cyclic voltammetry (CV) curves, (**b**) galvanostatic discharge-charge profiles measured under a current density of 100 mA g^−1^, (**c**) cycling performance at 100 mA g^−1^ and (**d**) rate capability.

**Figure 4 nanomaterials-09-01323-f004:**
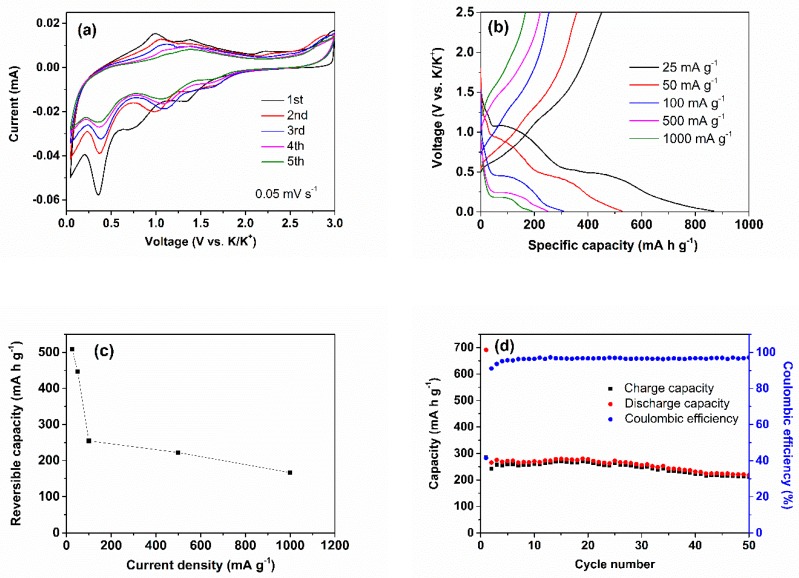
Electrochemical performance of the Sb_2_S_3_/CNT anode for K-ion batteries (KIBs). (**a**) CV curves, (**b**) the galvanostatic discharge-charge curves at various current densities, (**c**) rate capability and (**d**) cycling performance at 500 mA g^−1^.

**Table 1 nanomaterials-09-01323-t001:** Comparison of the electrochemical performance of the amorphous Sb_2_S_3_/CNT nanocomposites in this work, and those previously reported for Sb_2_S_3_-based anode materials for NIBs.

Anode Materials	Initial Coulomb Efficiency	Current Density (mA g^−1^)	Charge Capacity (mAh g^−1^)	Rate Capability (mAhg^−1^/mAg^−1^)	Voltage Cange (V)
Amorphous Sb_2_S_3_ [19]	65%	50	647	534/3000	0.01–2.5
Sb_2_S_3_@PPy [21]	63.7%	100	860	290/2000	0.01–3.0
MWNTs@Sb_2_S_3_@PPy [25]	75%	50	626	376/2000	0–2.0
Sb_2_S_3_-graphite [26]	84%	100	733	631/3000	0.01–3.0
Sb_2_S_3_/graphene [29]	65%	50	660	240/1500	0.01–2.0
HS Sb_2_S_3_/C [30]	64.8%	200	693	220/3200	0.01–3.0
Sb_2_S_3_/PPy [31]	70%	100	605	236/800	0.01–2.5
SN-rGO/Sb_2_S_3_ [32]	57%	100	592	365/2000	0.01–2.0
Sb_2_S_3_/SCS [33]	68.8%	100	642.8	263/1000	0.01–2.5
Sb_2_S_3_ HMS [34]	62%	200	616	314/3000	0.01–2.0
RGO/Sb_2_S_3_ nanorods [35]	52.6%	100	673	381/2000	0.01–2.0
Sb_2_S_3_/C [36]	78%	50	642	520/2000	0.005–2.0
Multi-shell Sb_2_S_3_ [37]	55%	100	901	604/2000	0.01–2.0
Amorphous Sb_2_S_3_/CNT (this work)	77.8%	100	870	441/3000	0.01–1.5

**Table 2 nanomaterials-09-01323-t002:** Comparison of the electrochemical performances of the amorphous Sb_2_S_3_/CNT nanocomposites in this work, and those previously reported for Sb_2_S_3_-based anodes for KIBs.

Anode Materials	Charge Capacity (mA h g^−1^/mA g^−1^)	Cycling Performance (mA h g^−1^/n)	Rate Capability (mA h g^−1^/mA g^−1^)	Voltage Range (V)	Ref.
Sb_2_S_3_@PPy coaxial nanorods	628/100	487/18	690/100 280/1000	0.01–3.0	[21]
Sb_2_S_3_-SNG composite	537/100	480/100	548/25 340/1000	0.1–3.0	[23]
Amorphous Sb_2_S_3_/CNT	286.5/500	212.4/50	451/25 166.6/1000	0.01–2.5	this work

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
