# Peer review of "Amorphous Sb2S3 Nanospheres In-Situ Grown on Carbon Nanotubes: Anodes for NIBs and KIBs"

_nanomaterials, 2019, doi:10.3390/nano9091323_

Round 1

Reviewer 1 Report

The authors present a study of the electrochemical performance for Sb2S3 nanoparticles grown on carbon nanotubes for use in sodium or potassium ion batteries. The sodium case is more promising than the potassium case. However both sets of results are interesting, particularly because of the performance improvements that are seen.

The work is well-motivated and the results are useful for use in new battery technologies. The manuscript is suitable for publication in Nanomaterials after the comments below are considered.

MAJOR COMMENTS

It is unclear why the testing parameters for the galvanostatic charge-discharge cycles were different for the NIB and the KIB testing. For example, why does Figure 3(b) show multiple cycles at a current density of 100 mA/g, while the corresponding Figure 4(b) shows one cycle at a current density of 25 mA/g? Similar inconsistencies also exist in the Supplementary material in Figures S2 and S3. Are there comparative data for pure Sb2S3 for KIBs corresponding to the results in Figure S2(a)? If so, they would be useful to include. The specifics of the electrochemical reactions occurring at the electrodes for the NIB and KIB cases are inconsistently described and not as clear as they could be. In the KIB case in particular, the description in Lines 181-198 is fairly redundant, and could be made more concise and clear. I do think separating the reactions out on individual lines (as in the KIB case, lines 195-197) is a good choice, which could be replicated, although see Minor Comment 1, below, about the formatting errors which exist there (among other places). A good addition would be a second table comparing results from the literature for the KIB case, as in Table 1 for the NIB case.

MINOR COMMENTS

The entire manuscript contains many formatting errors. For example, there are out-of-context Chinese characters in line 31. Additionally, most superscript and subscript formatting is missing in the manuscript after the abstract, which makes comprehension much more difficult. The figure part referred to for the SAED pattern at the end of the caption of Figure 2 (line 123) should be (d), not (b).

Author Response

Dear Editor and Reviewer 1:

Thank you for your letter and for the reviewers’ comments concerning our manuscript entitled “Amorphous Sb2S3 nanospheres in-situ grown on carbon nanotubes as anodes for NIBs and KIBs” Those comments are all valuable and very helpful for revising and improving our paper, as well as the important guiding significance to our researches.

Upon seriously taking the reviewers’ suggestions into account, we have tried our best to revise this manuscript accordingly. All the relevant changes are marked in red in the revised manuscript. The following is our response to the comments.

Thank you again.

Yours sincerely,

Jianqiu Deng

Comments and Suggestions for Authors

The authors present a study of the electrochemical performance for Sb2S3 nanoparticles grown on carbon nanotubes for use in sodium or potassium ion batteries. The sodium case is more promising than the potassium case. However both sets of results are interesting, particularly because of the performance improvements that are seen.

The work is well-motivated and the results are useful for use in new battery technologies. The manuscript is suitable for publication in Nanomaterials after the comments below are considered.

MAJOR COMMENTS

Q1: It is unclear why the testing parameters for the galvanostatic charge-discharge cycles were different for the NIB and the KIB testing. For example, why does Figure 3(b) show multiple cycles at a current density of 100 mA/g, while the corresponding Figure 4(b) shows one cycle at a current density of 25 mA/g? Similar inconsistencies also exist in the Supplementary material in Figures S2 and S3.

Response: Very good questions. We revised the Figure 4(b) and added the discharge-charge curves at various current densities.

For the electrochemical performance of the Sb2S3/CNTs anode for KIBs, we first evaluated the rate performance by a single discharge-charge cycle testing at various current densities. According to the rate performance, then we further tested the cycle performance at 500 mA g-1 because of the relative high capacities of the Sb2S3/CNTs anode under this current density. So this experiment scheme leads to different testing parameters of the galvanostatic charge-discharge testing for the NIBs and KIBs.

Q2: Are there comparative data for pure Sb2S3 for KIBs corresponding to the results in Figure S2(a)? If so, they would be useful to include.

Response: A good suggestion. I think it’s a pity that we have not evaluated the performance of pure Sb2S3 as an anode for KIBs. According the inferior performance of pure Sb2S3 anode for NIBs in this work, we speculated its performance is poor in KIBs.

Q3: The specifics of the electrochemical reactions occurring at the electrodes for the NIB and KIB cases are inconsistently described and not as clear as they could be. In the KIB case in particular, the description in Lines 181-198 is fairly redundant, and could be made more concise and clear. I do think separating the reactions out on individual lines (as in the KIB case, lines 195-197) is a good choice, which could be replicated, although see Minor Comment 1, below, about the formatting errors which exist there (among other places).

Response: A very good suggestion. In the revised manuscript, we have restated the electrochemical reactions of the electrode for the KIB and corrected the formatting errors.

Q4: A good addition would be a second table comparing results from the literature for the KIB case, as in Table 1 for the NIB case.

Response: we have compared the performance of the Sb2S3/CNTs anode for KIBs with reported results from the literature, which is listed in Table 2.

MINOR COMMENTS

Q5: The entire manuscript contains many formatting errors. For example, there are out-of-context Chinese characters in line 31. Additionally, most superscript and subscript formatting is missing in the manuscript after the abstract, which makes comprehension much more difficult. The figure part referred to for the SAED pattern at the end of the caption of Figure 2 (line 123) should be (d), not (b).

Response: Very good suggestions. We have corrected the formatting errors and other errors.

Reviewer 2 Report

The authors present relatively high electrochemical performance of Sb2S3/CNTs comparable to the previous reports, such as, a charge capacity of 870 mAh/g and coulomb efficiency of 77.8 % at 100 mA/g. Based on the potential of Sb2S3/CNTs, the authors demonstrate the electrochemical performance for K ions. The result includes some information for developing high capacity K-ion batteries. However, the discussion and interpretation are presented without clear evidence.

My major concerns are shown as follows;

1.      One of the issue is to solve structural expansion through intercalation in this paper, and the authors adopted amorphous Sb2S3 nanospheres on CNTs for this purpose. The structural information of the composite and its components is very important. However, there is no information of CNTs. The authors should show details of fabrication method and structural properties in the main text.

2.      In the section 3.2, the authors present relatively high electrochemical performance of Sb2S3/CNTs comparable to the previous reports, and concluded that the high performance can be attributed to structural integrity of Sb2S3/CNTs. However, there is no structural information during sodiation/desodiation. How do the authors confirm the structural integrity of Sb2S3/CNTs? The discussion on structural volume during redox reaction without structural information is also seen at the line #215 - #217 and #223 - #225 in the section 3.3.

In addition, the manuscript is not friendly to the readers as follows;

1.      At line #51, abbreviations of PPy and SNG are used without full spelling.

2.      At line #123, “SAED pattern is illustrated in b” would be “SAED pattern is illustrated in the inset of d”.

3.      Descriptions including English, including the font style, should be carefully improved for friendly to the readers. There are some mechanical errors.

From above reasons, the authors are advised to revise the manuscript carefully for further consideration of publication in nanomaterials.

Author Response

Dear Editor and Reviewer 2:

Thank you for your letter and for the reviewers’ comments concerning our manuscript entitled “Amorphous Sb2S3 nanospheres in-situ grown on carbon nanotubes as anodes for NIBs and KIBs” Those comments are all valuable and very helpful for revising and improving our paper, as well as the important guiding significance to our researches.

Upon seriously taking the reviewers’ suggestions into account, we have tried our best to revise this manuscript accordingly. All the relevant changes are marked in red in the revised manuscript. The following is our response to the comments.

Thank you again.

Yours sincerely,

Jianqiu Deng

Comments and Suggestions for Authors

The authors present relatively high electrochemical performance of Sb2S3/CNTs comparable to the previous reports, such as, a charge capacity of 870 mAh/g and coulomb efficiency of 77.8 % at 100 mA/g. Based on the potential of Sb2S3/CNTs, the authors demonstrate the electrochemical performance for K ions. The result includes some information for developing high capacity K-ion batteries. However, the discussion and interpretation are presented without clear evidence.

My major concerns are shown as follows;

Q1. One of the issue is to solve structural expansion through intercalation in this paper, and the authors adopted amorphous Sb2S3 nanospheres on CNTs for this purpose. The structural information of the composite and its components is very important. However, there is no information of CNTs. The authors should show details of fabrication method and structural properties in the main text.

Response: A very good suggestion. We have added the TEM images of CNTs in Figure S2, and revised the manuscript. The CNTs were purchased from XFNANO company (Nanjing, China).

From TEM image (Figure S2a), the CNTs show a wire structure and interconnect to one another. The diameters are in range of 20-90 nm. HRTEM image (Figure S2b) further reveals the amorphous carbon microstructure of the CNTs that are composed of short-range ordered nano-crystallites.

Q2. In the section 3.2, the authors present relatively high electrochemical performance of Sb2S3/CNTs comparable to the previous reports, and concluded that the high performance can be attributed to structural integrity of Sb2S3/CNTs. However, there is no structural information during sodiation/desodiation. How do the authors confirm the structural integrity of Sb2S3/CNTs? The discussion on structural volume during redox reaction without structural information is also seen at the line #215 - #217 and #223 - #225 in the section 3.3.

Response: Very good questions. I think it’s a pity that we have not tested the morphology and structure of the cycled Sb2S3/CNTs anodes for NIBs and KIBs. To verify the structural integrity of Sb2S3/CNTs, we cited the results reported in Refs. [19, 22, 25, 26] to support our statement. Especially, no obvious morphology differences of the pristine and cycled amorphous Sb2S3-graphite electrodes have been observed by using SEM (Ref. [26]), confirming the structure stability of amorphous Sb2S3 during cycling.

Q3. In addition, the manuscript is not friendly to the readers as follows;

At line #51, abbreviations of PPy and SNG are used without full spelling. At line #123, “SAED pattern is illustrated in b” would be “SAED pattern is illustrated in the inset of d”. Descriptions including English, including the font style, should be carefully improved for friendly to the readers. There are some mechanical errors.

Response: Good suggestions. We have revised and corrected the errors in the revised manuscript.

Round 2

Reviewer 2 Report

The authors present relatively high electrochemical performance of Sb2S3/CNTs comparable to the previous reports, such as, a charge capacity of 870 mAh/g and coulomb efficiency of 77.8 % at 100 mA/g. Based on the potential of Sb2S3/CNTs, the authors demonstrate the electrochemical performance for K ions. The result includes some information for developing high capacity K-ion batteries. All of concerns by the referee at the first review are responded, and the manuscript was well revised. From above reasons, the referee recommend the manuscript published as is.